# Chemosensory Characteristics of Brandies from Chinese Core Production Area and First Insights into Their Differences from Cognac

**DOI:** 10.3390/foods13010027

**Published:** 2023-12-20

**Authors:** Yue Ma, Yuanyi Li, Baochun Zhang, Chunhua Shen, Lina Yu, Yan Xu, Ke Tang

**Affiliations:** 1Lab of Brewing Microbiology and Applied Enzymology, School of Biotechnology, Jiangnan University, 1800 Lihu Avenue, Wuxi 214122, China; yuema@jiangnan.edu.cn (Y.M.); liyuanyiina@163.com (Y.L.); yxu@jiangnan.edu.cn (Y.X.); 2ChangYu Group Company Ltd., Yantai 264000, China; zbc01020304@163.com (B.Z.); zhang_bc@126.com (C.S.); zhj010203@163.com (L.Y.)

**Keywords:** aroma, cognac, odorants, flavor, partial least squares

## Abstract

This work aimed to compare the aroma characteristics of representative brandies with different grades from Yantai (one of the Chinese core production areas) and Cognac and to establish relationships between sensory descriptors and chemical composition. Descriptive analysis was performed with a trained panel to obtain the sensory profiles. Forty-three aroma-active compounds were quantified by four different methodologies. A prediction model on the basis of partial least squares analysis was performed to identify candidate compounds that were unique to a certain group of brandies. The result showed that brandies from Yantai could be distinguished from Cognac brandies on the basis of spicy, dried fruit, floral, and fruity-like aromas, which were associated with an aromatic balance between concentrations of a set of compounds such as 5-methylfurfural, *γ*-nonalactone, and *γ*-dodecalactone. Meanwhile, brandy with different grades could be distinguished on the basis of compounds derived mostly during the aging process.

## 1. Introduction

Brandy is commonly manufactured by distilling a wine prepared from white grapes followed by aging the distillate in oak barrels [1]. Factors involved in the multi-step process of brandy production include the type of grapes used, the fermentation process, vinification techniques, distillation, and the maturation period, and each step has a profound effect on the outcome of the overall profile and characteristics of the product [2]. Many countries in the world can produce brandy, but Cognac, whose name can only be used for brandies from a defined region in France, is considered one of the most famous brandies, and they are also synonymous with high quality. Brandy is also made across the new world, such as in South Africa and China. In recent years, China has had great potential for developing the brandy market; thus, understanding the differences between domestic brandy and other high-quality brandy and continuously optimizing the quality of domestic products is essential to capturing the consumer market. In China, Yantai located in Shandong Province is the most famous region, occupying one of the top places for market share by both volume and value [3]. Brandies are usually aged in oak barrels and labeled according to their age, and the main labeling grade in the brandy market both in Cognac and in Yantai includes VS (very special), VSOP (very special old pale), and XO (extra old). This classification generally relies on a system of well-known acronyms that refer to the age of the youngest component in the blend [2], but the age designations used in Cognac and Yantai are not exactly the same. The main difference is reflected in the definition of the XO grade, which means that the brandy has been stored in oak barrels at least for ten years in Cognac but six years in Yantai.

Considering the factors influencing consumers’ appreciation of different brandies, aroma is one of the drivers. In previous studies, using liquid–liquid extraction (LLE) and gas chromatography coupled to mass spectrometry (GC-MS), a total of 169 volatile compounds were directly identified in freshly distilled Cognac and Calvados [4]. However, it is widely accepted that the typical aroma is caused only by the odor-active compounds, which can be screened among the huge body of volatiles extracted and can be further localized by gas chromatography–olfactometry (GC-O) through human “sniffing” detection [5,6,7]. The odor-active compounds of brandies from various countries or regions, such as the brandies from Cognac [8], other regions in France [9], Slovakia [10], Peru [11], Portugal [12], and Germany [1], have been screened by GC-O analysis combined with GC-MS analysis. Previous results showed that the overall aroma of Cognac could be mimicked by an aroma recombinant consisting of 34 odorants [1]. In addition, the contribution of some compounds to the overall aroma of brandy was also proven, and these compounds belong to the families of terpenoids [13], lactones [14], and furans [15]. Composition and perception differences between brandies according to their origin have also been investigated. Results showed that some specific compounds could be related to the origin of brandies [16], but differences between these brandies were more likely due to an aromatic balance between concentrations of a set of common molecules rather than the presence of specific compounds with unique aromas [9]. Other results showed that the balance between concentrations of the pair (E)-*β*-damascenone and ethyl pentanoate could differentiate various brandies from Germany, France, and Spain [1]. Compared to other countries, the odor-active compounds relating to distinguishing brandies from China and its core production area remain to be investigated.

To explore this question, both sensory evaluation based on a trained expert panel and instrumental analysis based on an accurate quantitative analysis of odor-active compounds are necessary to obtain the chemosensory characteristics of brandies from target brandy production regions. Due to the differences between the physical and chemical properties of each odor-active compound and its content in samples, appropriate extraction, derivatization and detection techniques such as solvent-assisted flavor evaporation (SAFE), LLE, and solid-phase extraction (SPE) combined with GC-MS or comprehensive two-dimensional gas chromatography and time-of-flight mass spectrometry (GC × GC-TOFMS) should be selected for accurate chemical analysis [17,18]. Then, multivariate analysis can be used to explore the relationship between aroma attributes and volatile compounds and identify the candidate compounds that were unique to a certain group of brandies and their aroma attributes. Among multivariate techniques, partial least squares (PLS) regression has been used by several researchers for determining relationships between instrumental data (X-matrix) and descriptive sensory data (Y-matrix) [19], and PLS discriminant analysis (DA) has been used to identify those indicators that distinguish between different sample groups [20].

The objectives of the present study were the following: (1) to study the aroma attributes and odor active compounds of different brandies with different grades from one Chinese core production area and Cognac using sensory descriptive analysis and GC-O analysis, respectively; (2) to accurately quantify odor-active compounds by multiple quantitative methods and to compare the differences of these compounds between different brandies; (3) to elucidate the relationship between aroma attributes and odor active compounds by PLS and to identify the compounds that could potentially contribute to odor perception relating to different groups of brandies.

## 2. Materials and Methods

### 2.1. Brandy Samples

Commercial brandy samples from different regions were provided by ChangYu Winery (Changyu Group Co., Ltd., Yantai, China). These brandies were selected by brandy experts from the China Alcoholic Drinks Association to ensure they were typical of the brandy styles of the production area. All samples were stored horizontally at 20 °C in the dark before being used. The information about these samples was provided by ChangYu Winery and is given in Table 1.

### 2.2. Chemicals

Absolute ethanol (≥99.8%, GC grade), dichloromethane (≥99.8%, GC grade), diethyl ether (≥99.8%, GC grade), and methanol (≥99.9%, GC grade) were purchased from Sigma-Aldrich Trading Co., Ltd. (Shanghai, China). Ultrapure water was obtained from a Milli-Q purification system (Millipore, Bedford, MA, USA). Analytical-grade anhydrous sodium sulfate (Na_2_SO_4_), sodium chloride (NaCl), and sodium bicarbonate (NaHCO_3_) were purchased from China National Pharmaceutical Group Corp. (Shanghai, China). All standards used for quantification in the experiment (see Appendix A) were chromatographic grade with purity ≥97% and were obtained from Sigma-Aldrich Trading Co., Ltd. (Shanghai, China).

### 2.3. Sample Preparation

Liquid–Liquid extraction (LLE) was used to extract volatile compounds for gas chromatography–olfactometry (GC–O) and gas chromatography–mass spectrometric (GC–MS) analysis. A 25 mL sample of brandy was diluted with 75 mL of saturated sodium chloride solution and the sample was placed in a separatory funnel. For quantification, the sample was spiked with 10 μL of 4-methyl-2-pentanol (12,010 mg·L^−1^), 20 μL of menthol (5090 mg·L^−1^), 400 μL of methyl hexanoate (91.8 mg·L^−1^) and 10 μL of pivalic acid (11,970 mg·L^−1^), which were internal standards before being extracted. The diluted sample was extracted three times with 50 mL of redistilled diethyl ether. The combined extracts were further dried with anhydrous Na_2_SO_4_ and concentrated to a final volume of 500 μL under a gentle stream of nitrogen, and then stored at −20 °C until analysis. All extractions were carried out in triplicate on two experimental replicates.

### 2.4. Gas Chromatography-Olfactometry (GC-O) and Gas Chromatography-Mass Spectrometric (GC-MS) Analysis

GC-O and GC-MS analysis was conducted on an Agilent 6890 gas chromatograph equipped with an Agilent 5975 mass-selective detector and a sniffing port (ODP 2, Gerstel, Germany). The analytical column was a DB-FFAP column (60 m × 0.25 mm i.d., 0.25 μm film thickness, Agilent, Torrance, CA, USA). The front inlet was programmed in splitless mode (1 uL injected), and the oven temperature was held at 50 °C for 2 min, increased to 230 °C at 5 °C min^−1^, and then held at 230 °C for 15 min. Helium was used as a carrier gas at a constant flow rate of 2 mL min^−1^, and the column effluent split to the mass-selective detector and sniffing port was 1:1. The temperature of the injector, transfer line, and olfactory port were set at 240 °C. The data acquisition (electron impact (EI) at 70 eV) for the mass-selective detector was in scan mode, 35–500 Da for compound identification.

GC-O analysis was conducted on the DB-FFAP column by two well-trained assessors from the Laboratory of Brewing Microbiology and Applied Enzymology at Jiangnan University. The two assessors underwent six months of at least twice-weekly training in aroma identification and intensity differentiation, and their performance in detecting and identifying different odor qualities was evaluated by using a procedure similar to the European Test of Olfactory Capabilities (ETOC) [21]. The method used in GC-O analysis was time intensity [22]. During each sniff, assessors were asked to detect the presence of an odor by recording the retention time as soon as they perceived it and then gave a descriptor of the perceived odor and rated its intensity from 1.0 to 5.0 (in which 3.0 stood for medium, and 5.0 stood for the highest). The analysis was repeated in triplicate by each subject. If one odor zone was detected by both assessors at least once in the triplicate, the odor zone was retained, and the odor intensity of the odor zone was averaged.

The compound responsible for the odor zone was identified by (1) GC–MS and comparing the retention indices (RI) and odor descriptor of a candidate compound with the RI and odor descriptor of its pure standards under the same GC conditions as GC-O; and Kovats retention indices (RIs) of peaks were calculated by injection of a reference solution of *n*-alkanes (C_5_–C_30_) under the same GC conditions. (2) Comparing the odor descriptor of a candidate compound with its odor descriptor reported in the database. (3) Comparing the RI and MS spectrum of a candidate compound with its RI and MS spectrum reported in the National Institute of Standards and Technology (NIST) mass spectral library.

### 2.5. Quantification of Odor Active Compounds

Four methods were involved in the quantitation of odor-active compounds in brandy on the basis of an external standard calibration graph obtained by plotting the relative individual peak areas against the concentrations of the calibration standards. Calibration standards with different concentrations were diluted in brandy model solution which was prepared with EtOH/H_2_O (40:60, *v*/*v*), and data obtained from solutions with at least eight concentrations was subjected to regression analysis using the least squares method to construct an external calibration curve, and internal standards were used for normalization of the data. Three replicate injections of two experimental replicates were performed, and the results were averaged. The linearity, accuracy, and limit of quantification (LOQ) of each compound were checked. The information for the calibration curve of each compound is given in Appendix A.

For major compounds, namely ethyl acetate, 2-methyl-1-propanol, 3-methyl-1-butanol, and ethyl lactate, their quantification was achieved by direct injection using a GC- flame ionization detector (FID). The brandy sample was placed in a 2 mL vial and injected into the GC-FID at a split ratio of 1:10 with 1 μL each time. Heating-up procedure: hold at 50 °C for 3 min, ramp up to 110 °C at a rate of 6 °C·min^−1^, then ramp up to 200 °C at a rate of 30 °C·min^−1^, and hold for 1 min. Helium was used as the carrier gas at a flow rate of 1 mL·min^−1^ and hydrogen at a flow rate of 40 mL·min^−1^.

The quantification of 2,3-butanedione was achieved by using headspace solid-phase microextraction-gas chromatography-mass spectrometry after derivatization following a procedure modified from the one we conducted previously [17]. First, 2 mL of sample was diluted with 6 mL of saturated sodium chloride solution and the diluted sample was placed into a 20 mL glass, then spiked with 2 μL of *p*-fluorobenzaldehyde (4.103 mg·L^−1^), which was an internal standard, and 300 μL of O-(2,3,4,5,6-pentafluorobenzyl)hydroxylamine hydrochloride (PFBHA, 20 g·L^−1^). The mixture was equilibrated at 65 °C for 10 min and extracted for 45 min under stirring at the same temperature, then transferred the fiber to the injector for desorption at 250 °C for 300 s. The condition of the GC was the same as in 2.4. The mass spectrometer was operated in electron ionization mode at 70 eV with SIM. The ion monitored for *p*-fluorobenzaldehyde after derivatization was *m*/*z* 319. Monitored ions of 2,3-butanedione after derivatization were 279.

The quantification of (*E*)-whiskeylactone, (*Z*)-whiskeylactone, *γ*-nonalactone, and *γ*-dodecalactone was achieved by using stir bar sorptive extraction (SBSE) combined with comprehensive two-dimensional gas chromatography and time-of-flight mass spectrometry (GC × GC-TOFMS) following the procedure we conducted previously [14].

The quantification of other minor odor-active compounds in brandy was achieved by using LLE coupled with GC-MS as described before.

### 2.6. Assessors

Fifteen healthy assessors (between the ages of 18 and 25, 9 females and 6 males) were recruited and selected from sixty-seven healthy assessors at Jiangnan University. They went through screening tests that evaluated their performance in discriminating between different odor qualities and different odor intensity levels, as well as their performance in logic scaling, and they were also given a three-month (twice-a-week) basic sensory training on brandy sensory evaluation. All of the assessors confirmed that they were in good health, had normal olfactory abilities, and had neither smoking nor allergy histories. All assessors provided informed consent in line with the Helsinki Declaration, and they were instructed not to wear any perfume and not eat or drink anything for at least 1 h before the study. Assessors received gifts for their participation.

### 2.7. Procedure of Sensory Evaluation

Before the formal experiment, all assessors received six months of training (1–2 times per week) and these training sessions could be divided into three parts. In the first session part, all of the 54 aroma standards from Le Nez du Vin were provided to the assessors to help them memorize different aroma descriptors. And this session was repeated until all assessors were able to recognize and name the different aroma descriptors. In the second session part, different brandy samples were provided for assessors to become familiar with the aroma characteristics of brandies and then the assessors were asked to assess and describe brandy samples in terms of aroma characteristics. In this part, a list of aroma descriptors and their corresponding aroma references, which are given in Appendix A, were generated by discussing with the participants in order to prevent overlapping and redundancies among descriptors for describing different brandy samples. In the last session part, aroma references with different aroma intensities were provided for assessors to help them memorize and discriminate between different aroma intensity levels.

### 2.8. Quantitative Descriptive Analysis of Brandy Samples

Quantitative descriptive analysis was conducted by the selected and trained assessors in a standard sensory evaluation room with separated booths at room temperature (25 °C) to evaluate the aroma profile of nine brandy samples which were prepared in International Standards Organization glass (20 mL) for the orthonasal pathway. The brandy samples were prepared 10 min before the analysis and the glass was covered with tin foil paper to avoid as little aroma dispersion as possible. Each sample was coded by three random digits, and the order of presentation of the samples was counterbalanced. During the evaluation, the assessors were asked to use the list of descriptors previously generated to score the intensity of the aroma descriptors of the nine brandy samples on a numerical scale from 0 to 6, where 0 represented that a descriptor was barely perceived and 6 represented the highest intensity of a descriptor. Assessors were given a rest of 10 min between each sample. Two replicated experiments were conducted, and the results were averaged.

### 2.9. Data Analysis

The sensory panel results were subjected to analysis of variance (ANOVA, *p* = 95%) to verify the ability of the assessors with *R* software (version 4.0.1) by using the *panelperf* function from the *SensoMineR* package [23], in which the dependent variables were the eight aroma attributes and the independent variables were the factors associated with the sample effect, the subject effect, the session effect, and all their first-order interactions.

A one-way ANOVA was performed by SPSS Statistics (IBM, version 22) on the sensory data to investigate the samples’ differences in each aroma attribute. Differences between samples were considered statistically significant at *p*-value < 0.05.

A statistical analysis was performed on the concentration of each compound quantified in different samples with a one-way analysis of variance (ANOVA). When the differences were significant (*p* < 0.05), Duncan’s test was used to check the differences between pairs of samples and was carried out using SPSS Statistics (IBM, version 22).

Principal component analysis (PCA) was also carried out on the sensory data of the aroma attribute for brandy samples of different brands over their grades by using the *prcomp* function of the *tempR* package [24] on *R* software (version 4.0.1). The PCAs were used to provide a global representation of the trajectory of brandy samples of different brands in relation to the evolution of aroma attributes based on the first and second principal components. The categorized aroma attribute trajectories in each brandy brand are illustrated by connecting two grades (brandy with grade XO from Changyu included two samples).

Partial least squares-discriminant analysis (PLS-DA) was carried out on instrumental data of brandy samples grouped by core production area (Yantai and Cognac) and grade (VSOP and XO) by using the *splsda* function of the *mixOmics* package. Partial least squares regression (PLS-R) was performed with XLSTAT (version 2019) to correlate the odor-active compounds (X-matrix) and sensory data (Y-matrix) of brandy samples. All variables were mean-centered and normalized to unit variance before applying PLS analyses. Variable Importance on Projection (VIP) values were calculated to estimate the importance of each odor-active compound in the PLS projection, and the most important explanatory elements of the examined relationships were selected using the Variable Identification (VID) procedure (Ooms, 1996).

## 3. Results

### 3.1. Aroma Profile of Different Brandies

The ability of the assessors to evaluate the aroma profile of brandy samples was verified by ANOVA (*p* = 95%), and the results of the ANOVA (Appendix A) showed the panel’s ability to differentiate the aroma descriptor (discrimination) consistently (repeatability) and consensually (agreement). There is a significant sample effect on all descriptors, and it is possible to conclude that the assessors were able to differentiate and correctly identify different odor descriptors.

The aroma of nine samples of brandies from four brands with two grades (VSOP and XO) was described by the sensory panel using eight different descriptors, and a one-way ANOVA was performed to compare the differences in a single attribute between different samples. The result is given in Table 2. It shows that sample CVS was mainly characterized by odors of spicy and dried fruit; sample CX10 was mainly characterized by odors of spicy and toasted; sample CX15 was mainly characterized by toasted and dried fruit; sample HVS was mainly characterized by odors of alcohol and caramel; sample HXO was mainly characterized by odors of toasted and caramel; sample MVS was mainly characterized by odors of alcohol and caramel; sample MXO was mainly characterized by odors of caramel and spicy; sample RVS was mainly characterized by odors of alcohol and spicy; sample RXO was mainly characterized by odors of caramel and toasted.

Principal component analysis (PCA) was conducted to follow the aroma profile modification of brandy samples from different brands induced by the grade effect, and the PCA map is shown in Figure 1. The first two principal components of the PCA applied to the sensory data accounted for 69.93% of the total variance. As can be observed, a trend to discriminate between brandies from Yantai and Cognac samples along the PC1 is apparent (Figure 1). The samples from Yantai were located in the right part of the plot and were clearly separated from the Cognac samples, which were located in the left part.

It shows that the spicy and dried fruit attributes had the highest positive contribution to PC1 and floral and fruit had the highest negative contribution to PC1. It means that the samples from Yantai can be discriminated from the samples from Cognac based on their higher spicy and dried fruit aromas, and lower floral and fruit sensations. It also showed that the dried fruit and spicy attributes were negatively correlated with the floral and fruit attributes, and the alcohol and mushroom attributes were negatively correlated with the toasted attribute, which were separated along the PC2. The samples presented in the PCA map were marked as trajectories of brandy samples from VSOP grade to XO grade. It shows that brandies from Yantai obviously changed from the right bottom part to the right top part in Figure 1, which suggested that the aroma profile of higher grade Chinese brandies changed from the spicy to dried fruit attribute, and brandies from Cognac obviously changed from the left bottom part to the left top part in Figure 1, which suggested that the aroma profile of higher grade Cognac brandies changed from the attributes of alcohol and/or mushroom to fruit, floral and caramel [25].

### 3.2. Identification and Quantification of Odor-Active Compounds in Different Brandies

There were 51 odor zones detected by GC-O analysis across all brandy samples, and a number of sample pre-treatment methods combined with GC-FID, GC-MS, and GC × GC-TOFMS analysis were conducted to identify and quantify the odor-active compounds responsible for each OZ obtained in GC-O analysis. The result of the identification of odor-active compounds in different brandies is given in Appendix A. It shows that 13 esters, 8 alcohols, 6 acids, 5 phenols, 5 furans, 4 lactones, 2 aromatics, 1 terpene, 2 aldehydes, and ketones were associated with 46 odor zones, but 5 odor zones failed to be related to compounds identified by instrumental analysis, and these odor zones were carrying a toasted, smoky or spicy-like odor. The quantitative results of these odor-active compounds in different brandies are given in Table 3, except for the compounds methyl salicylate, creosol, and ethyl cinnamate, due to the unavailability of chemical standards.

Among all these odor-active compounds, 18 compounds met 2 conditions at the same time: (1) they were at least presented in one sample with a GC-O score not less than 3, and (2) all of them could be clearly perceived in all brandy samples (Appendix A). Representative compounds were 2-phenylethanol and acetic acid, which were presented in all of the samples with a GC-O score above 4, carrying a rose and acid-like odor. Among all these odor-active compounds, most of the compounds presented in brandies were at the mg·L^−1^ level, and a small number of compounds were at the μg·L^−1^ level. There were 4 compounds with concentrations above 100 mg·L^−1^, namely ethyl acetate, 3-methyl-1-butanol, 2-methyl-1-propanol, and acetic acid (Table 3).

A one-way ANOVA was performed on the concentration of each compound across different samples, and it shows that all of these odor-active compounds had significant differences between different brandy samples. Duncan’s test was further applied to check the differences between pairs of samples at a *p*-value equal to 0.05. The result is given in Table 3. In order to determine which compounds were more important in the discrimination of the different brandy samples, PLS-DA was carried out on the chemical data of the brandy samples which were grouped by core production area (Yantai and Cognac) and grade (VSOP and XO), and the resulting map is given in Figure 2. Three quality indices are reported in the PLS-DA model: R^2^X (cum), R^2^Y (cum), and Q^2^(cum). R^2^X and R^2^Y represent the model interpretation rate, and Q^2^ indicates the model predictive ability; R^2^Y and Q^2^ closer to 1 indicate that the model is more stable and reliable. Figure 2a shows that the PLS-DA score plots separated brandies from Yantai and Cognac well, and the values of R^2^X, R^2^Y, and Q^2^ were 0.340, 0.959, and 0.843. Figure 2c showed that the PLS-DA score plots separated brandies from VSOP and XO not as well as the former, since the values of R^2^X, R^2^Y, and Q^2^ were 0.256, 0.910, and 0.596 in components 1 of the PLS-DA model. Figure 2b,d provided the most important odor-active compounds in discriminating brandy samples from different regions and grades based on component 1 of the PLS-DA model. It shows (Figure 2b) that 5-methylfurfural, γ-nonalactone, ethyl 2-hydroxy-4-methylpentanoate, ethyl lactate, and γ-dodecalactone were the most important in discriminating brandy samples from Yantai and Cognac, while ethyl vanillate, ethyl acetate, 2-acetylfuran, vanillin, and acetic acid had a greater “importance” in discriminating brandy samples from VSOP and XO.

### 3.3. Correlation between Odor-Active Compounds and Aroma Attributes of Brandies

Partial least squares regression analysis (PLS-R) was used to explore the relationship between the concentration of odor-active compounds and aroma attributes in different brandies. The correlation loadings of 43 odor-active compounds (X-variables) and 8 aroma attributes (Y-variables) on the first two components are shown in Figure 3. It shows that 54.8% of odor-active compounds data variation explained 69.8% of aroma attribute data variation, but the values of Q^2^ were 0.388, which suggested that the predictability of the 8 aroma attributes on the basis of the 43 odor-active compounds is not well. In a PLS-R loading plot, if variables appear on the circumference of the circle, they are highly correlated and appear close to each other in the case of positive correlations, while they appear far apart in the case of negative correlations. It can be seen that the attributes “caramel”, “mushroom”, and “alcohol” were closer to zero, which were not well represented on the first two components; thereby, they should be poorly connected with the 43 odor-active compounds. It shows that “fruity” and “floral” were positively correlated with each other, and “spicy” and “dried fruit” were negatively correlated with “fruity” and “floral”. As part of the PLS-R analyses, a VIP was performed for each explanatory variable to show which variable contributed most to the model, and the result is given in Appendix A. Compounds with VIP > 1 are the most relevant for explaining sensory data, and it shows that there were 24 compounds for the first two explanatory variables contributing to the prediction, and compounds 5-methylfurfural, γ-nonalactone, ethyl lactate, and γ-dodecalactone were the most relevant for explaining sensory data on the first two explanatory variables. VID is defined as the correlation coefficients between the original X variables and the Y variables predicted by the PLS-R model, and it was performed to select the most important odor-active compounds related to each aroma attribute, and compounds with VID coefficients higher than 0.8 in absolute value for each attribute are given in Appendix A. For the “alcohol” attribute, no compounds had significant positive regression coefficients, but 2-acetylfuran and 2-methyl-1-propanol showed significant negative coefficients; for the “dried fruit” attribute, (Z)-whiskeylactone, 5-methylfurfural, ethyl acetate, and ethyl lactate had significant positive regression coefficients; for the “floral” and “fruity” attribute, γ-dodecalactone and γ-nonalactone had significant positive regression coefficients, and ethyl lactate showed significant negative coefficients for both attributes, while 5-methylfurfural and furfural had significant negative coefficients only for “fruity” attribute. On the contrary, for the “spicy” attribute, 1-hexanol, 5-methylfurfural, ethyl lactate, and furfural had significant positive regression coefficients, while γ-dodecalactone and γ-nonalactone showed significant negative coefficients; for the “toasted” attribute, 2-acetylfuran, ethyl acetate, ethyl vanillate, and vanillin had significant positive regression coefficients; for the “mushroom” attribute, benzaldehyde and β-damascenone had significant positive regression coefficients. For the “caramel” attribute, it shows that no compounds had significant positive or negative regression coefficients.

## 4. Discussion

Descriptive sensory analysis has been performed on the evaluation of the aroma profiles of various brandies. Lexicons including butter, bouillon, box tree, hay, grass, pear, pepper, rose, and lime tree have been used to describe freshly distilled cognac [8]; vanilla, dry wood, green wood, brioche, leather, and tobacco have been used to describe Cognac eaux-de-vie aged in barrels [26]; and fruity, malty, baked apple, honey, clove, flowery, vanilla, and coconut have been used to describe commercial cognac [1]. In our result (Figure 1), eight aroma attributes have been agreed upon in a preliminary session to describe the brandies from Yantai and Cognac, and it shows that brandies from Yantai had a relatively higher intensity on spicy and dried fruit, while brandies from Cognac had a relatively higher intensity on floral and fruity. Previous studies suggested that the aging system and aging time significantly influenced some attributes such as woody, caramel, green, fruity, and toasted [25]. Our results are consistent with that conclusion, which shows (Figure 1) that as the grade of brandy increased, the aroma profile of brandies from Yantai converted from spicy to dried fruit. Previous results [27] suggested that the main descriptors for cognacs belonging to the fruity, floral, spicy, and woody families for aging periods of 40 or more years, and the attribute of green was usually negatively correlated with the overall quality of brandies and with the age of brandies [28,29]. In our studies, alcohol and/or mushroom were selected to describe brandies instead of green, and our results showed that the aroma profile of Cognac brandies converted from the attributes of alcohol and/or mushroom to fruity, floral, and caramel, as the grade of brandy increased.

By performing GC-O and GC-MS analysis on our brandy samples, 2-phenylethanol, and acetic acid were revealed as the most intense compounds presented in all of the brandy samples detected by GC-O, while ethyl acetate, 3-methyl-1-butanol, 2-methyl-1-propanol, and acetic acid had the highest concentrations in the brandy samples. Acetic acid has a vinegar-like odor, and it normally occurs in spoiled wines, but during the aging process, its concentration could significantly increase [30]. Studies showed that high-quality Cognacs have a certain amount of acetic acid from 18.65 mg·L^−1^ to 454.9 mg·L^−1^ [31]. In our result (Table 3), the amount of acetic acid content ranged from 100 mg·L^−1^ to 150 mg·L^−1^, which is consistent with high-quality cognac levels. Higher alcohols such as 2-phenylethanol and 3-methyl-1-butanol were revealed as important odor-active compounds in these brandy samples with content above 100 mg·L^−1^. It is believed that higher alcohols generally make a modest contribution to the vinous nature of fermented beverages but are unlikely to serve as impact odorants [18,32]. The majority of the higher alcohols are formed as a byproduct of yeast amino acid metabolism [33]. Our result shows that ethyl acetate was the most prevalent ester and esters were the most prevalent odor-active compounds that could be perceived as having a sweet and fruity-like aroma. During the aging process, different organic acids slowly undergo esterification reactions with ethanol, and thus the ethyl ester content in brandy gradually increases [34]. Previous results [1] showed that in Hennessy Cognac, (*E*)-*β*-damascenone, which has a sweet fruit-like odor, had the highest flavor-dilution factor of 2048. In our result, this compound was only detected in four brandies from Cognac.

Studies on how to distinguish brandy samples from different regions and different aging times by particular compounds have been conducted in the past. Research showed that the pair of (*E*)-*β*-damascenone and ethyl pentanoate could be treated as indicators to differentiate various cognacs from German, French, and Spanish brandies [1]. For freshly distilled French brandies within a limited geographic area, it is also possible to distinguish different samples from an aromatic balance between concentrations of a set of common molecules such as ethyl esters and alcohols [9], as well as furanic compounds and aliphatic linear alcohols [16]. Among all of the odor-active compounds detected by GC-O in our research, 5-methylfurfural, *γ*-nonalactone, ethyl 2-hydroxy-4-methylpentanoate, ethyl lactate, and *γ*-dodecalactone were revealed to play the greatest role in discriminating brandy samples from Yantai and Cognac. The result obtained from PLS-R showed that *γ*-dodecalactone and *γ*-nonalactone had significant positive regression coefficients for “floral” and “fruity” attributes, and 5-methylfurfural had significant positive regression coefficients for “spicy” and “dried fruit” attributes. The conclusion should be based on the results obtained from descriptive analysis since brandy samples from Yantai and Cognac had different on floral, fruity, spicy, and dried fruit-like aromas. Differences in the signature compounds between our result and results from other research can be caused by differences in brandy samples or by the limitation in the usage of those compounds only detected by GC-O analysis. Brandy with high quality is often aged for a period of several decades, and it is also possible to predict brandy age accurately by a PLS model with a smaller subset consisting of ethyl esters and methyl ketones [35]. Our result showed that ethyl vanillate, ethyl acetate, 2-acetylfuran, vanillin, and acetic acid had a greater impact in discriminating brandy samples from VSOP and XO, which had differences in aging years. Most of these compounds were oak extracts, which have the most typical aromatic characteristics that distinguish brandy from wine and usually provide smoky, toasty, spicy, and other aromas. Previous studies showed that not only the aging system but also the aging time could modify the flavor of brandies [25]. It is reasonable for these compounds to be treated as predictors to distinguish brandies with different grades since physical processes can influence the extract compounds that are derived mostly or entirely from oak during the aging process [36].

## 5. Conclusions

In this study, the aroma profile and odor-active compounds of different brandies with different qualities from Yantai and Cognac were studied by descriptive analysis and GC-O combined with GC-FID, GC-MS, and GC × GC-TOFMS analysis. A total of 43 compounds were quantitated by four methods on the basis of a standard calibration graph. Results of sensory data and instrumental data showed that brandy from Yantai had a relatively higher intensity on spicy and dried fruit, while brandies from Cognac had a relatively higher intensity on floral and fruity, and the differences in their aroma characteristics could be associated with an aromatic balance between concentrations of a set of compounds such as 5-methylfurfural, *γ*-nonalactone, ethyl 2-hydroxy-4-methylpentanoate, ethyl lactate, and *γ*-dodecalactone. Meanwhile, VSOP brandy could be distinguished from XO brandy on the basis of compounds derived mostly during the aging process, such as ethyl vanillate, ethyl acetate, 2-acetylfuran, vanillin, and acetic acid.

## Figures and Tables

**Figure 1 foods-13-00027-f001:**
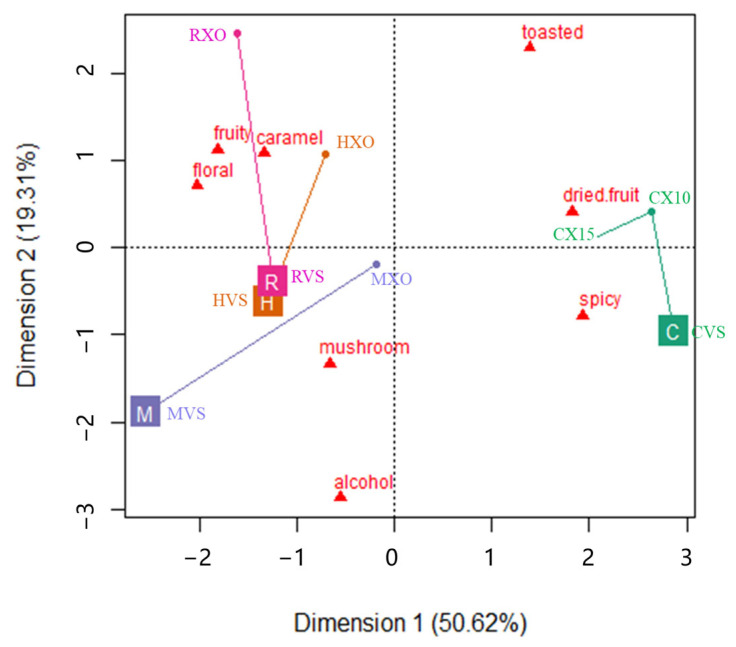
The first two dimensions of the PCA biplot show the aroma attribute trajectories of brandy samples from different brands (symbolized as C, H, M, and R) over two grades (VSOP and XO). The beginning of the trajectory was VSOP (position at the brand marker), and the end of the trajectory was XO (position at the solid dots).

**Figure 2 foods-13-00027-f002:**
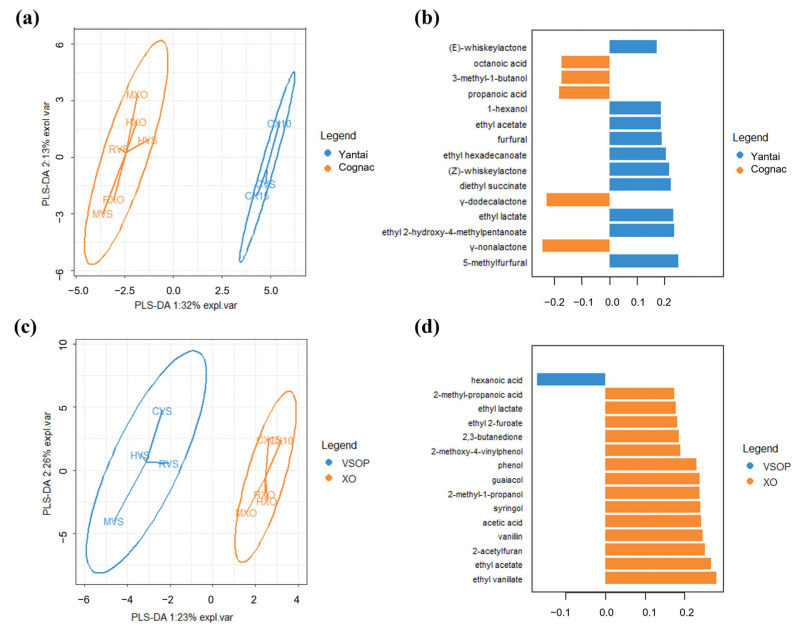
A plot of PLS-DA was carried out on the chemical data of brandy samples, which were grouped by region (Yantai and Cognac) and grade (VSOP and XO). (**a**) Sample plots of the chemical data after a basic PLS-DA model were operated on this data, and the samples were depicted with the confidence ellipses of region class labels, and plots used the first two components as axes. (**b**) Loading plot from the PLS-DA applied to the chemical data to discriminate brandy samples from different regions based on component 1 of the PLS-DA model. The colors indicated the odor-active compounds, with the median being the maximum for each region. The most relevant odor-active compounds (those with the greatest absolute loading value) were at the bottom of the plot, and only the top 15 important variables were shown in the plot. (**c**) Sample plots of the chemical data after a basic PLS-DA model were operated on this data, and the samples were depicted with the confidence ellipses of grade class labels, and plots used the first two components as axes. (**d**) Loading plot from the PLS-DA applied to the chemical data to discriminate brandy samples from different grades based on component 1 of the PLS-DA model. Colors indicated the odor-active compounds, with the median being the maximum for each grade.

**Figure 3 foods-13-00027-f003:**
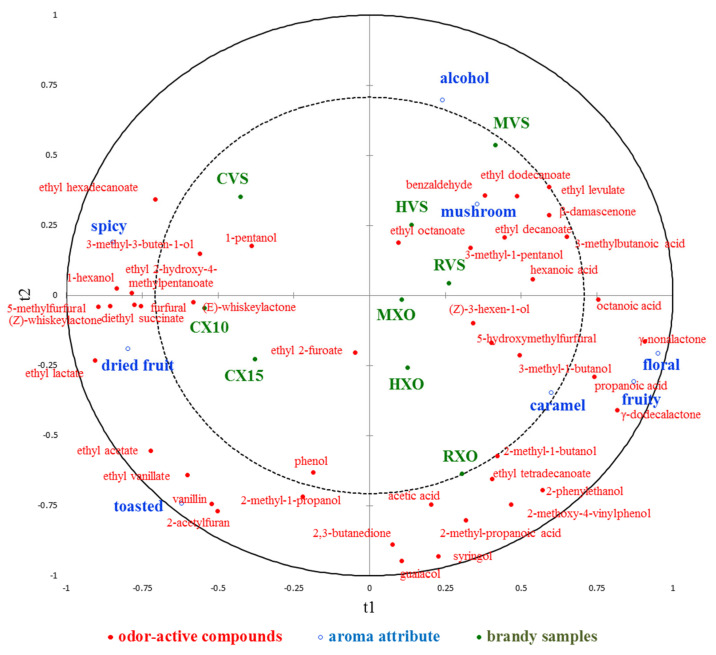
The correlation loadings of 43 volatile compounds (X-variables) and 8 aroma attributes (Y-variables) on the first 2 components of the PLS-R model.

**Table 1 foods-13-00027-t001:** Information about brandy samples.

Sample Code	Grade ^1^	Region	Grape Cultivar	Region of Oak Barrels	Alcohol% (*v*/*v*)
CVS	VSOP	Yantai, China	Ugni Blanc	Limousin, France	40
CX10	XO10	Yantai, China	Ugni Blanc	Limousin, France	40
CX15	XO15	Yantai, China	Ugni Blanc	Limousin, France	40
HVS	VSOP	Cognac, France	Ugni Blanc, Colombard, Folle Blanche	Limousin, France	40
HXO	XO	Cognac, France	Ugni Blanc, Colombard, Folle Blanche	Limousin, France	40
MVS	VSOP	Cognac, France	Ugni Blanc, Colombard, Folle Blanche	Troncais, France	40
MXO	XO	Cognac, France	Ugni Blanc, Colombard, Folle Blanche	Troncais, France	40
RVS	VSOP	Cognac, France	Ugni Blanc, Colombard, Folle Blanche	Limousin, France	40
RXO	XO	Cognac, France	Ugni Blanc, Colombard, Folle Blanche	Limousin, France	40

^1^ “VSOP” means that the brandy has been stored in oak barrels for at least four years; “XO” means that the brandy has been stored in oak barrels for at least ten years; and “XO10” or “XO15” represents that the brandy has been stored in oak barrels for ten or fifteen years, respectively.

**Table 2 foods-13-00027-t002:** Intensity of brandy aroma attributes evaluated by descriptive analysis.

Sample	CVS	CX10	CX15	HVS	HXO	MVS	MXO	RVS	RXO
alcohol	4.00 ± 0.26 ^d^	3.33 ± 0.15 ^b^	4.80 ± 0.29 ^ef^	4.50 ± 0.16 ^e^	3.50 ± 0.12 ^bc^	5.60 ± 0.39 ^g^	3.80 ± 0.20 ^cd^	5.00 ± 0.21 ^f^	2.80 ± 0.25 ^a^
dried fruit	5.03 ± 0.15 f^g^	4.80 ± 0.40 ^f^	5.20 ± 0.38 ^g^	2.50 ± 0.14 ^a^	3.00 ± 0.21 ^b^	3.20 ± 0.27 ^bc^	3.80 ± 0.23 ^de^	3.50 ± 0.19 ^cd^	4.00 ± 0.16 ^e^
floral	0.85 ± 0.26 ^a^	1.00 ± 0.25 ^a^	1.50 ± 0.24 ^b^	2.50 ± 0.07 ^c^	2.50 ± 0.12 ^c^	3.50 ± 0.14 ^d^	2.50 ± 0.12 ^c^	3.50 ± 0.19 ^d^	4.00 ± 0.24 ^e^
fruity	1.01 ± 0.14 ^a^	1.00 ± 0.28 ^a^	1.20 ± 0.23 ^a^	2.00 ± 0.39 ^b^	2.60 ± 0.12 ^c^	2.90 ± 0.26 ^c^	2.00 ± 0.19 ^b^	3.80 ± 0.19 ^d^	4.00 ± 0.21 ^d^
caramel	1.47 ± 0.23 ^a^	3.00 ± 0.38 ^c^	3.70 ± 0.29 ^d^	3.80 ± 0.19 ^d^	4.50 ± 0.19 ^e^	4.80 ± 0.19 ^ef^	5.30 ± 0.28 ^g^	2.50 ± 0.25 ^b^	5.00 ± 0.14 ^fg^
toasted	3.98 ± 0.18 ^cd^	5.00 ± 0.16 ^g^	5.50 ± 0.41 ^h^	3.00 ± 0.27 ^b^	4.50 ± 0.31 ^ef^	2.50 ± 0.25 ^a^	4.20 ± 0.21 ^de^	3.80 ± 0.16 ^c^	4.60 ± 0.31 ^f^
spicy	5.52 ± 0.40 ^e^	5.30 ± 0.47 ^de^	4.80 ± 0.29 ^c^	3.60 ± 0.22 ^ab^	3.90 ± 0.38 ^ab^	3.80 ± 0.21 ^ab^	5.00 ± 0.19 ^cd^	4.00 ± 0.16 ^b^	3.50 ± 0.07 ^a^
mushroom	0.98 ± 0.22 ^bc^	1.18 ± 0.29 ^c^	0.70 ± 0.30 ^b^	0.38 ± 0.28 ^a^	1.00 ± 0.16 ^bc^	3.50 ± 0.12 ^d^	4.00 ± 0.34 ^e^	1.00 ± 0.20 ^bc^	1.00 ± 0.23 ^bc^

Statistical analysis was performed on the intensity of each attribute perceived in different samples with a one-way analysis of variance (ANOVA). When the differences were significant (*p* < 0.05), Duncan’s test was used to check the differences between pairs of samples, and the samples sharing the same letters were not different at *p* = 0.05.

**Table 3 foods-13-00027-t003:** Quantification of odor-active compounds detected by GC-O and GC-MS in brandies from Yantai and Cognac.

No.	Compounds	Samples								
		CVS	CX10	CX15	HVS	HXO	MVS	MXO	RVS	RXO
Concentration (mg L^−1^)										
1	ethyl acetate	488.3 ± 2.8 ^b^	807.3 ± 5.6 ^d^	814.3 ± 23.1 ^d^	409.2 ± 30.3 ^a^	583.5 ± 11.9 ^c^	366.0 ± 11.1 ^a^	512.4 ± 16.9 ^b^	376.7 ± 28.3 ^a^	584.3 ± 28.8 ^c^
3	2-methyl-1-propanol	226.3 ± 10.1 ^b^	393.2 ± 3.3 ^de^	336.9 ± 9.8 ^c^	331.0 ± 27.7 ^c^	396.2 ± 3.2 ^de^	97.2 ± 3.1 ^a^	409.7 ± 6.5 ^e^	332.0 ± 3.2 ^c^	360.8 ± 3.0 ^cd^
4	2-methyl-1-butanol	30.7 ± 0.9 ^a^	38.2 ± 1.2 ^ab^	42.0 ± 3.3 ^abc^	39.8 ± 2.1 ^abc^	56.7 ± 5.7 ^d^	35.3 ± 2.0 ^ab^	50.5 ± 6.7 ^cd^	51.1 ± 1.5 ^cd^	45.2 ± 3.6 ^bcd^
5	3-methyl-1-butanol	562.5 ± 13.4 ^a^	679.5 ± 2.7 ^cde^	622.2 ± 8.8 ^b^	669.6 ± 27.1 ^cd^	709.0 ± 4.0 ^e^	684.8 ± 6.2 ^de^	710.0 ± 4.7 ^e^	642.3 ± 3.7 ^bc^	664.8 ± 2.9 ^cd^
9	1-hexanol	17.0 ± 1.0 ^c^	31.6 ± 3.1 ^d^	10.9 ± 0.3 ^b^	4.6 ± 0.4 ^a^	11.1 ± 1.4 ^b^	6.5 ± 0.2 ^a^	15.5 ± 1.1 ^c^	5.7 ± 0.7 ^a^	5.7 ± 0.2 ^a^
10	ethyl lactate	10.6 ± 0.7 ^d^	20.5 ± 0.2 ^f^	15.8 ± 0.8 ^e^	6.3 ± 0.8 ^b^	8.4 ± 0.2 ^c^	3.9 ± 0.2 ^a^	8.1 ± 0.2 ^c^	5.5 ± 0.1 ^b^	6.9 ± 0.1 ^bc^
11	(Z)-3-hexen-1-ol	0.8 ± 0.1 ^ab^	1.1 ± 0.1 ^bc^	0.7 ± 0.1 ^a^	0.9 ± 0.1 ^ab^	1.3 ± 0.1 ^c^	1.0 ± 0.0 ^b^	1.7 ± 0.1 ^d^	1.0 ± 0.1 ^b^	1.0 ± 0.1 ^b^
13	acetic acid	101.8 ± 3.2 ^a^	129.8 ± 1.5 ^de^	130.3 ± 2.5 ^de^	121.1 ± 1.7 ^bc^	127.5 ± 2.9 ^cde^	119.2 ± 0.9 ^b^	126.2 ± 2.5 ^bcde^	124.9 ± 2.6 ^bcd^	133.7 ± 0.5 ^e^
14	furfural	9.0 ± 0.8 ^d^	13.0 ± 1.2 ^e^	8.6 ± 0.4 ^cd^	5.2 ± 0.4 ^b^	9.0 ± 0.5 ^d^	0.0 ± 0.0 ^a^	7.1 ± 0.3 ^c^	0.0 ± 0.0 ^a^	0.0 ± 0.0 ^a^
18	propanoic acid	0.0 ± 0.0 ^a^	0.0 ± 0.0 ^a^	0.0 ± 0.0 ^a^	2.1 ± 0.1 ^b^	0.0 ± 0.0 ^a^	3.9 ± 0.2 ^c^	4.2 ± 0.6 ^c^	3.9 ± 0.6 ^c^	7.8 ± 1.1 ^d^
19	2-methyl-propanoic acid	1.9 ± 0.2 ^a^	2.2 ± 0.1 ^a^	2.9 ± 0.3 ^bc^	2.2 ± 0.2 ^a^	2.5 ± 0.2 ^abc^	2.3 ± 0.1 ^ab^	2.3 ± 0.1 ^ab^	2.6 ± 0.2 ^abc^	3.1 ± 0.2 ^c^
20	5-methylfurfural	2.2 ± 0.2 ^d^	2.2 ± 0.0 ^cd^	2.0 ± 0.3 ^cd^	1.0 ± 0.1 ^a^	1.7 ± 0.2 ^bc^	1.2 ± 0.1 ^a^	1.1 ± 0.1 ^a^	1.3 ± 0.1 ^ab^	1.2 ± 0.2 ^ab^
21	ethyl levulate	0.0 ± 0.0 ^a^	0.0 ± 0.0 ^a^	0.7 ± 0.1 ^bc^	0.4 ± 0.1 ^b^	1.0 ± 0.2 ^c^	9.2 ± 0.3 ^e^	1.0 ± 0.1 ^c^	1.0 ± 0.1 ^c^	2.4 ± 0.1 ^d^
22	ethyl decanoate	1.3 ± 0.5 ^bc^	0.8 ± 0.1 ^ab^	0.2 ± 0.0 ^a^	0.2 ± 0.0 ^a^	3.0 ± 0.0 ^d^	3.1 ± 0.1 ^d^	1.4 ± 0.2 ^c^	0.8 ± 0.0 ^ab^	1.1 ± 0.1 ^bc^
24	3-methylbutanoic acid	0.9 ± 0.1 ^bc^	1.2 ± 0.1 ^b^	1.1 ± 0.1 ^bc^	1.7 ± 0.2 ^c^	1.2 ± 0.1 ^b^	2.4 ± 0.3 ^d^	0.6 ± 0.0 ^a^	1.8 ± 0.2 ^c^	1.8 ± 0.2 ^c^
25	diethyl succinate	4.8 ± 0.5 ^d^	3.2 ± 0.1 ^c^	3.2 ± 0.1 ^c^	2.0 ± 0.1 ^a^	2.3 ± 0.1 ^ab^	1.8 ± 0.0 ^a^	2.2 ± 0.2 ^a^	2.0 ± 0.1 ^a^	2.9 ± 0.1 ^bc^
28	hexanoic acid	2.5 ± 0.2 ^b^	1.6 ± 0.2 ^a^	2.4 ± 0.1 ^b^	2.5 ± 0.0 ^b^	2.5 ± 0.0 ^b^	2.6 ± 0.0 ^b^	1.9 ± 0.0 ^a^	2.6 ± 0.1 ^b^	2.6 ± 0.1 ^b^
29	guaiacol	0.5 ± 0.1 ^b^	0.5 ± 0.0 ^b^	1.4 ± 0.2 ^d^	0.6 ± 0.1 ^bc^	1.4 ± 0.2 ^d^	0.1 ± 0.0 ^a^	1.0 ± 0.2 ^c^	0.7 ± 0.0 ^bc^	2.1 ± 0.2 ^e^
31	2-phenylethanol	6.6 ± 0.7 ^a^	6.3 ± 0.4 ^a^	7.2 ± 0.1 ^a^	10.4 ± 0.7 ^ab^	13.5 ± 0.8 ^bc^	8.3 ± 0.1 ^a^	16.6 ± 2.3 ^c^	14.9 ± 1.6 ^c^	31.9 ± 2.4 ^d^
35	octanoic acid	10.8 ± 1.6 ^abc^	8.6 ± 0.8 ^ab^	6.6 ± 0.7 ^a^	9.7 ± 1.5 ^abc^	19.0 ± 2.3 ^d^	25.7 ± 0.3 ^e^	11.6 ± 1.5 ^bc^	13.6 ± 1.8 ^c^	21.9 ± 0.9 ^de^
39	syringol	0.4 ± 0.0 ^ab^	0.6 ± 0.0 ^bc^	1.4 ± 0.1 ^d^	0.9 ± 0.1 ^c^	1.9 ± 0.2 ^e^	0.1 ± 0.0 ^a^	1.4 ± 0.1 ^d^	0.8 ± 0.1 ^bc^	2.5 ± 0.3 ^f^
41	5-hydroxymethylfurfural	23.8 ± 1.7 ^a^	30.6 ± 2.8 ^a^	32.0 ± 1.3 ^a^	34.2 ± 2.0 ^a^	69.4 ± 5.0 ^c^	30.3 ± 0.8 ^a^	45.8 ± 0.1 ^b^	89.5 ± 7.8 ^d^	31.9 ± 2.7 ^a^
42	vanillin	3.8 ± 0.4 ^b^	5.0 ± 0.3 ^cd^	5.8 ± 0.1 ^d^	2.0 ± 0.1 ^a^	4.4 ± 0.2 ^bc^	2.1 ± 0.1 ^a^	2.5 ± 0.2 ^a^	2.5 ± 0.1 ^a^	5.5 ± 0.7 ^d^
Concentration (ug L^−1^)										
2	2,3-butanedione	8.9 ± 0.4 ^a^	17.1 ± 1.2 ^c^	20.5 ± 2.2 ^c^	9.5 ± 0.2 ^a^	16.3 ± 1.6 ^bc^	8.8 ± 0.7 ^a^	11.4 ± 0.8 ^ab^	20.0 ± 0.9 ^c^	29.0 ± 3.4 ^d^
6	1-pentanol	117.4 ± 14.1 ^bc^	526.5 ± 71.8 ^e^	0.0 ± 0.0 ^a^	0.0 ± 0.0 ^a^	0.0 ± 0.0 ^a^	156.9 ± 20.1 ^c^	402.3 ± 30.6 ^d^	55.4 ± 8.1 ^ab^	0.0 ± 0.0 ^a^
7	3-methyl-3-buten-1-ol	162.6 ± 3.7 ^b^	543.4 ± 20.5 ^c^	0.0 ± 0.0 ^a^	27.4 ± 3.3 ^a^	150.2 ± 5.9 ^b^	136.7 ± 11.8 ^b^	145.1 ± 20.9 ^b^	0.0 ± 0.0	0.0 ± 0.0
8	3-methyl-1-pentanol	249.9 ± 32.4 ^cd^	259.2 ± 12.9 ^cd^	0.0 ± 0.0 ^a^	182.0 ± 13.8 ^b^	341.4 ± 8.8 ^e^	257.2 ± 4.6 ^cd^	349.6 ± 14.3 ^e^	278.7 ± 38.0 ^d^	201.8 ± 2.6 ^bc^
12	ethyl octanoate	125.7 ± 28.0 ^b^	274.8 ± 9.4 ^c^	16.8 ± 1.9 ^a^	45.3 ± 6.3 ^ab^	394.9 ± 44.5 ^d^	320.4 ± 19.1 ^cd^	337.8 ± 54.0 ^cd^	53.6 ± 8.3 ^ab^	40.6 ± 5.8 ^ab^
15	2-acetylfuran	265.4 ± 30.0 ^bc^	338.1 ± 9.4 ^d^	289.2 ± 7.2 ^bcd^	166.7 ± 3.9 ^a^	301.6 ± 19.2 ^bcd^	158.7 ± 11.4 ^a^	244.7 ± 17.1 ^b^	250.4 ± 21.2 ^b^	324.6 ± 40.6 ^cd^
16	benzaldehyde	0.0 ± 0.0 ^a^	0.0 ± 0.0 ^a^	0.0 ± 0.0 ^a^	0.0 ± 0.0 ^a^	0.0 ± 0.0 ^a^	91.7 ± 4.4 ^b^	118.1 ± 2.3 ^c^	0.0 ± 0.0 ^a^	0.0 ± 0.0 ^a^
17	ethyl 2-hydroxy-4-methylpentanoate	350.0 ± 47.3 ^d^	279.9 ± 12.6 ^cd^	291.4 ± 3.9 ^cd^	250.2 ± 9.3 ^bc^	191.9 ± 25.8 ^ab^	172.6 ± 8.5 ^a^	194.4 ± 27.1 ^ab^	179.5 ± 9.5 ^ab^	237.0 ± 13.7 ^abc^
23	ethyl 2-furoate	51.5 ± 8.0 ^bc^	60.4 ± 0.7 ^c^	43.1 ± 5.8 ^ab^	30.5 ± 2.1 ^a^	82.6 ± 2.1 ^d^	49.1 ± 1.9 ^bc^	75.7 ± 1.5 ^d^	47.6 ± 4.1 ^bc^	44.9 ± 4.5 ^b^
26	*β*-damascenone	0.0 ± 0.0 ^a^	0.0 ± 0.0 ^a^	0.0 ± 0.0 ^a^	0.0 ± 0.0 ^a^	344.1 ± 2.1 ^d^	421.2 ± 54.0 ^e^	278.6 ± 6.9 ^c^	159.4 ± 14.6 ^b^	0.0 ± 0.0 ^a^
27	ethyl dodecanoate	453.5 ± 17.7 ^de^	352.0 ± 26.7 ^cd^	0.0 ± 0.0 ^a^	153.0 ± 21.5 ^ab^	599.9 ± 11.6 ^e^	1187.7 ± 113.2 ^f^	393.1 ± 48.1 ^cd^	244.2 ± 12.1 ^bc^	518.5 ± 68.7 ^de^
30	(*E*)-whiskeylactone	538.9 ± 46.2 ^c^	707.4 ± 43.5 ^e^	882.2 ± 10.7 ^f^	360.7 ± 12.7 ^a^	540.8 ± 16.2 ^c^	607.5 ± 1.9 ^cd^	668.3 ± 2.5 ^de^	444.2 ± 3.3 ^b^	399.2 ± 1.7 ^ab^
32	(*Z*)-whiskeylactone	698.0 ± 36.8 ^c^	860.0 ± 36.2 ^d^	1050.5 ± 9.9 ^e^	469.0 ± 6.3 ^a^	634.2 ± 15.5 ^b^	638.0 ± 5.6 ^bc^	678.6 ± 1.9 ^bc^	515.1 ± 2.9 ^a^	465.7 ± 1.4 ^a^
33	phenol	53.7 ± 5.4 ^abcd^	51.9 ± 5.6 ^abc^	73.2 ± 9.3 ^cd^	35.4 ± 4.2 ^a^	61.0 ± 8.4 ^bcd^	53.0 ± 6.7 ^abcd^	50.6 ± 4.4 ^ab^	34.6 ± 3.7 ^a^	74.2 ± 6.4 ^d^
34	ethyl tetradecanoate	0.0 ± 0.0 ^a^	0.0 ± 0.0 ^a^	0.0 ± 0.0 ^a^	0.0 ± 0.0 ^a^	0.0 ± 0.0 ^a^	0.0 ± 0.0 ^a^	0.0 ± 0.0 ^a^	25.8 ± 2.1 ^b^	87.2 ± 9.5 ^c^
36	*γ*-nonalactone	26.4 ± 1.2 ^b^	20.5 ± 1.2 ^a^	25.8 ± 2.2 ^ab^	43.5 ± 1.1 ^d^	47.0 ± 1.0 ^d^	52.9 ± 0.3 ^e^	45.9 ± 4.1 ^d^	35.8 ± 0.5 ^c^	57.5 ± 0.7 ^e^
37	2-methoxy-4-vinylphenol	0.0 ± 0.0 ^a^	0.0 ± 0.0 ^a^	0.0 ± 0.0 ^a^	0.0 ± 0.0 ^a^	206.7 ± 10.8 ^c^	0.0 ± 0.0 ^a^	166.3 ± 11.1 ^b^	61.0 ± 7.1 ^a^	278.2 ± 28.7 ^d^
38	ethyl hexadecanoate	294.1 ± 16.0 ^c^	203.2 ± 70.0 ^b^	58.1 ± 7.5 ^a^	36.1 ± 3.5 ^a^	39.9 ± 0.1 ^a^	83.5 ± 12.0 ^a^	16.2 ± 0.4 ^a^	20.2 ± 2.4 ^a^	52.3 ± 10.8 ^a^
40	*γ*-dodecalactone	16.3 ± 1.6 ^a^	26.8 ± 1.2 ^b^	31.9 ± 1.2 ^c^	36.6 ± 1.0 ^de^	40.8 ± 2.3 ^ef^	37.2 ± 1.6 ^def^	34.0 ± 1.0 ^cd^	40.2 ± 0.6 ^ef^	41.7 ± 0.6 ^f^
43	ethyl vanillate	149.4 ± 11.4 ^bcd^	255.5 ± 31.3 ^f^	203.1 ± 25.7 ^def^	93.0 ± 12.8 ^ab^	235.2 ± 19.6 ^ef^	74.0 ± 6.7 ^a^	143.2 ± 5.6 ^bc^	91.3 ± 10.0 ^ab^	185.2 ± 11.5 ^cde^

Note: Statistical analysis was performed on the concentration of each compound quantified in different samples with a one-way analysis of variance (ANOVA). When the differences were significant (*p* < 0.05), Duncan’s test was used to check the differences between pairs of samples, and the samples sharing the same letters were not different at *p* = 0.05.

## Data Availability

Data is contained within the article or Appendix A.

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
