# Peer review of "Chemosensory Characteristics of Brandies from Chinese Core Production Area and First Insights into Their Differences from Cognac"

_foods, 2023, doi:10.3390/foods13010027_

Round 1

Reviewer 1 Report

Comments and Suggestions for Authors

The work is interesting and has an analytical component that was arduous and well done.

However, the major problem is related to doubts about the experimental design and corresponding analysis of variance. The authors analyzed 9 commercial samples (3 from China and 6 from France from the Cognac region) with different categories regarding aging time. It is not clear how many experimental replicates were used to estimate the experimental error when performing the one-way ANOVA.

On the other hand, it does not make sense to carry out two-way ANOVA, given that these are commercial samples in which there is no guarantee that the initial distillate (before aging) is the same, so the variability deduced from the factor aging time includes the variability of factors that are not controlled (harvest year, fermentation, distillation aging conditions...etc..).

As a result of this experimental design, the writing style must be reviewed and changed, namely in the title, abstract and conclusions, since the writing style used suggest that the 3 samples from China and the 6 from France represent the entire production of the respective country/region, and this from a sampling and statistical point of view raises serious questions.

Additionally, it is suggested to replace the word brandy with aged wine spirit, given that in some countries they are synonymous but in others they correspond to different products.

Line 34-38 - The meaning of the age designations used in the Cognac region and in China should be better explained, as these designations do not always have the same meaning and the regulations of each region should be consulted.

Line 42-44 - These quantitative data must have an accessible reference that can be consulted

Line 53-59 – I suggest to add two references with GCO results in wine spirits:

Caldeira I., Bruno de Sousa R., Belchior A. P., ClímacoM. C. 2008 A sensory and chemical approach to the aroma of wooden aged Lourinhã wine brandy. Ciência Tec. Vitiv 23(2) 97-110  http://scielo.pt/scielo.php?script=sci_arttext&pid=S0254-02232008000200003&lng=pt&tlng=en.

Rogerson, F.S.S. & de Freitas, V.A.P., 2002. Fortification Spirit, a Contributor to the Aroma Complexity of Port. J.Food Sci. 67(4): 1564-1569.doi:10.1111/j.1365-2621.2002.tb10323.x 

Line 99 – The information of samples in the Table 1 is collected from bottles label? This must be clarified.

Line 129 –“The analytical coumns were DB….”..I think the author intend to write “ The analytical column was DB….” Check the sentence please

Line 139 – It would be interesting to furnish sensory test results concerning their ability in discrimination and detection odors.

Line 207 – It should be explained what odors standards were used since le nez du vin has several different kits  

Line 231 -  I think  the aroma is evaluated only by orthonasal pathway. It is ok but I suggest to explain better this aspect.

Line 242 – The same doubt reported before. The author repeated analyzing another bottle of the same sample or they repeated the analysis of the same sample? Based on the text it seems the authors used the last option and in that case the variability correspond to method variability and not to the experimental variability. This matter must be well explained and clarified

Figure 1 - It is not clear why the authors do not project all samples onto the two-dimensional PCA plane. It will be important to do this to better follow the discussion of the results

Line 290-293 – see the third paragraph of the revision.

Line 326 – Could the high intensity of the mushroom descriptor in the MVS and MXO samples indicate a sensory defect in these samples? Because this descriptor is not common in aged spirits (see Leauté et al 1998 or Caldeira et al., 2017). This must be discussed.

Léauté R., Mosedale J.R., Mourgues J., Puech J.-L., 1998. Barrique et vieillissement des eaux-de-vie. In: Oenologie fondements scientifiques et technologiques. 1085-1142. Flanzy C. (ed.), Collection Sci.& Tech. Agr.a, New York. (cited by Lurton et al 2012)

Caldeira I., Anjos O., Belchior A.P., Canas S. 2017 Sensory impact of alternative ageing technology for the producing of wine brandies. Ciência Téc. Vitiv. 32(1) 12-22 DOI: https://doi.org/10.1051/ctv/20173201012

Line 326 - Instead or region I suggest to replace by provenance singe China is not a region

Line 395 – concerning the results of odor active compounds must be clarified what results are used. The concentration or its intensity from GCO results?

Line 457 and 470 – Improve the writing because it seems the reference [24] has been studied the extraction of acetic acid from the wood or the formation of ethyl acetate and this reference only summarize the results of other teams that studied that subjects

Line 480-Check the reference that is not in the final list

Line 496 and lines 512-514- Weren't the differences between VSOP and XO more than expected considering all the vast knowledge about aging in wood? This aspect should be included in the discussion

Verify the references list in accordance with journal rules

Author Response

Reviewer #1

The work is interesting and has an analytical component that was arduous and well done.

However, the major problem is related to doubts about the experimental design and corresponding analysis of variance. The authors analyzed 9 commercial samples (3 from China and 6 from France from the Cognac region) with different categories regarding aging time. It is not clear how many experimental replicates were used to estimate the experimental error when performing the one-way ANOVA.

Response: We are sorry that the text was misleading. We used two bottles of the same sample in our experiment, and for the instrumental analysis, extraction and measurement were carried out in triplicate for each bottle, and the result of each bottle was averaged. For the sensorial analysis, two replicate experiments were conducted. The explanation for each part has been added at Line 131, Line 173 and Line 239.

On the other hand, it does not make sense to carry out two-way ANOVA, given that these are commercial samples in which there is no guarantee that the initial distillate (before aging) is the same, so the variability deduced from the factor aging time includes the variability of factors that are not controlled (harvest year, fermentation, distillation aging conditions...etc..).

Response: Thank you for the comments, after consideration and checking, we agree with the reviewer, and the analysis and result associated with this part has been removed from the article.

As a result of this experimental design, the writing style must be reviewed and changed, namely in the title, abstract and conclusions, since the writing style used suggest that the 3 samples from China and the 6 from France represent the entire production of the respective country/region, and this from a sampling and statistical point of view raises serious questions.

Response: We agree with the reviewer’s comment. In the parts of the title, abstract, conclusions, and other related text, the words “China” and “France” have been replaced by “Yantai” or “one Chinese core production area” and “Cognac”.

Additionally, it is suggested to replace the word brandy with aged wine spirit, given that in some countries they are synonymous but in others they correspond to different products.

Response: Thank you for your suggestion. We prefer to use the word “brandy” because the definition of the brandy in this study was given in the first paragraph, and we do not think it could be misleading to use the word brandy in this article.

Line 34-38 - The meaning of the age designations used in the Cognac region and in China should be better explained, as these designations do not always have the same meaning and the regulations of each region should be consulted.

Response:  Thank you for the suggestion. The explanation has been added at Lines 44-51 as following, “Brandies are usually aged in oak barrels and labeled according to their age, and the main labeling grade in the brandy market both in Cognac and in Yantai includes VS (very special), VSOP (very special old pale) and XO (extra old). This classification generally relies on a system of well-known acronyms that refer to the age of the youngest com-ponent in the blend [2], but the age designations used in Cognac and in Yantai are not exactly the same. The main difference is reflected in the definition of the XO grade, which means that the brandy has been stored in oak barrels at least for ten years in Cognac but six years in Yantai.”

Line 42-44 - These quantitative data must have an accessible reference that can be consulted

Response: We can not find an accessible online reference to support this quantitative data, so we replace the sentence with this sentence: “n China, Yantai located in Shandong Province is the most famous region, occupying one of the top places for market share by both volume and value, and ChangYu Winery is the leading company in this production area [3].”  at Lines 41-43.

Line 53-59 – I suggest to add two references with GCO results in wine spirits:

Caldeira I., Bruno de Sousa R., Belchior A. P., ClímacoM. C. 2008 A sensory and chemical approach to the aroma of wooden aged Lourinhã wine brandy. Ciência Tec. Vitiv 23(2) 97-110  http://scielo.pt/scielo.php?script=sci_arttext&pid=S0254-02232008000200003&lng=pt&tlng=en.

Rogerson, F.S.S. & de Freitas, V.A.P., 2002. Fortification Spirit, a Contributor to the Aroma Complexity of Port. J.Food Sci. 67(4): 1564-1569. . doi:10.1111/j.1365-2621.2002.tb10323.x 

Response:  After a careful reading of the two literatures, we decided to cite the first article. The second reference is not selected because we would like the article to be more related to the strict definition of brandy, rather than all the wine spirits. The new reference has been added at Line 61.

Line 99 – The information of samples in the Table 1 is collected from bottles label? This must be clarified.

Response: It has been clarified and the sentence is “The information about these samples was provided by ChangYu Winery and is given in Table 1.” at Lines 105-106.

Line 129 –“The analytical coumns were DB….”..I think the author intend to write “ The analytical column was DB….” Check the sentence please

Response: Thank you for pointing it out. The sentence has been revised to “The analytical column was a DB-FFAP column” at Line 137.

Line 139 – It would be interesting to furnish sensory test results concerning their ability in discrimination and detection odors.

Response: Thank you for the suggestion. A sentence, “and their performance in detecting and identifying different odor qualities was evaluated by using a procedure similar to the European Test of Olfactory Capabilities (ETOC)[21].”, has been added at Lines 148-150  to state their ability in discrimination and detection of odors.

Line 207 – It should be explained what odors standards were used since le nez du vin has several different kits  

Response: At the first step, all of the 54 aroma standards from Le Nez du Vin were provided to the assessors to help them memorize different aroma descriptors, and after discussion, the following odor standards (almond, mushroom, rose, blackcurrant, caramel, cinnamon, clove, toast, smoky) from LE NEZ DU VIN were used. The sentence has been revised to “In the first session part, all of the 54 aroma standards from Le Nez du Vin were provided to the assessors to help them memorize different aroma descriptors.” at Lines 215-217.

Line 231 -  I think  the aroma is evaluated only by orthonasal pathway. It is ok but I suggest to explain better this aspect.

Response: Yes, thank you, this point has been clarified at Line 231.

Line 242 – The same doubt reported before. The author repeated analyzing another bottle of the same sample or they repeated the analysis of the same sample? Based on the text it seems the authors used the last option and in that case the variability correspond to method variability and not to the experimental variability. This matter must be well explained and clarified.

Response: We are sorry that the text was misleading. We used another bottle of the same sample in our experiment, so two replicate experiments were conducted and the results were averaged. The explanation has been added at Line 239.

Figure 1 - It is not clear why the authors do not project all samples onto the two-dimensional PCA plane. It will be important to do this to better follow the discussion of the results.

Response: We are sorry that Figure 1 was not clearly labeled. All samples were projected onto the two-dimensional PCA plane before, but they were not labeled in the figure. The figure has been revised.

Line 290-293 – see the third paragraph of the revision.

Response: After consideration and checking, we agree with the reviewer that it was not reasonable to carry out two-way ANOVA since there was no guarantee that the initial distillate was the same. This part has been removed from the article, and other relevant sections have also been revised.

Line 326 – Could the high intensity of the mushroom descriptor in the MVS and MXO samples indicate a sensory defect in these samples? Because this descriptor is not common in aged spirits (see Leauté et al 1998 or Caldeira et al., 2017). This must be discussed.

Léauté R., Mosedale J.R., Mourgues J., Puech J.-L., 1998. Barrique et vieillissement des eaux-de-vie. In: Oenologie fondements scientifiques et technologiques. 1085-1142. Flanzy C. (ed.), Collection Sci.& Tech. Agr.a, New York. (cited by Lurton et al 2012)

Caldeira I., Anjos O., Belchior A.P., Canas S. 2017 Sensory impact of alternative ageing technology for the producing of wine brandies. Ciência Téc. Vitiv. 32(1) 12-22 DOI: https://doi.org/10.1051/ctv/20173201012

Response:  Thank you for the suggestion, we read the two articles and the discussion related to this point has been added at Lines 444-455. The discussion part is:

Previous studies suggested that aging system and aging time significantly influenced some attributes such as woody, caramel, green, fruity and toasted [25]. Our results are consistent with that conclusion, which show (Figure 1) that as the grade of brandy in-creased, the aroma profile of Chinese brandies converted from spicy to dried fruit. Previous results [27] suggested that the main descriptors for cognacs belonging to the fruity, floral, spicy and woody families for aging periods of 40 or more years, and the attribute of green was usually negatively correlated with the overall quality of brandies and with the age of brandies [28, 29]. In our studies, alcohol and/or mushroom were selected to describe brandies instead of green, and our results showed that the aroma profile of Cognac brandies converted from attribute alcohol and/or mushroom to fruity, floral and caramel as the grade of brandy increased.

Line 326 - Instead or region I suggest to replace by provenance singe China is not a region

Response: According to the reviewers’ suggestion, China has been replaced by “Yantai”, and the word “region” in some parts of the article has been replaced by “Chinese core production area”.

Line 395 – concerning the results of odor active compounds must be clarified what results are used. The concentration or its intensity from GCO results?

Response: Thank you for the suggestion, the sentence has been clarified at Lines 392-394, “Partial least squares regression analysis (PLS-R) was used to explore the relation-ship between the concentration of odor-active compounds and aroma attributes in different brandies.”

Line 457 and 470 – Improve the writing because it seems the reference [24] has been studied the extraction of acetic acid from the wood or the formation of ethyl acetate and this reference only summarize the results of other teams that studied that subjects

Response: Thank you for the suggestion, the sentence has been revised and new reference has been added at Line 459-461.

The revised sentence is: “Acetic acid has a vinegar-like odor, and it normally occurs in spoiled wines, but during aging process, its concentration could significantly increase [30].”

Line 480-Check the reference that is not in the final list

Response: Thank you for the reminding, the reference has been added in the final list.

Line 496 and lines 512-514- Weren't the differences between VSOP and XO more than expected considering all the vast knowledge about aging in wood? This aspect should be included in the discussion

Response: Thank you for the suggestion, the sentence has been revised.

The revised sentences are:

Line 496 (Lines 502-506): Previous studies showed that not only the aging system but also the aging time could modify the flavor of brandies [25]. It is reasonable for these compounds to be treated as predictors to distinguish brandies with different grades since physical processes can influence the extract compounds that are derived mostly or entirely from oak during the aging process [36].

Line 512-514 (Lines 517-519): “Meanwhile, VSOP brandy could be distinguished from XO brandy on the basis of compounds derived mostly during the aging process, such as ethyl vanillate, ethyl ac-etate, 2-acetylfuran, vanillin, and acetic acid.”

Verify the references list in accordance with journal rules

Response: All of the references have been revised in accordance with the rules of Foods.

Reviewer 2 Report

Comments and Suggestions for Authors

Suggestion of title: Chemosensory characteristics of representative brandies from China and first insights into its differences from Cognacs

The aroma atributes and odor active compounds of different brandies are directly related to aging. Table 1 should provide the origins of the oak used in barrels of the different brandy samples studied.

2.6 subjects could be changed for assessors

Table 3. Revise the concentration of the compounds. Some of them are not in µg/L, as mentioned in lines 459-460, 462-463.

Conclusions:

Lines 509-510: the aromas should be associated with the origins of the brandies, as described in Discussion (lines 444-446)

LInes 511-512: the same approach shoud be used here. Associate the set of aromatic compounds with the brandies from China and Cognac, as described in Discussion (lines 480-484)

Lines 512-514: once again the same approach shoud be used here. Associate the compounds derived from oak to VSOP and XO cognacs, as described in Discussion (lines 494-496) and showed in Figure 2d.

Author Response

Reviewer #2

Suggestion of title: Chemosensory characteristics of representative brandies from China and first insights into its differences from Cognacs

Response: Thank you for the suggestion. Considering suggestions from all reviewers, the title has been revised to “Chemosensory characteristics of brandies from Chinese core production area and first insights into their differences from Cognac”.

The aroma attributes and odor active compounds of different brandies are directly related to aging. Table 1 should provide the origins of the oak used in barrels of the different brandy samples studied.

Response: The origins of the oak used in the barrels of the different brandy samples have been added to Table 1.

2.6 subjects could be changed for assessors

Response: all of “subjects” in the text have been changed for “assessors”

Table 3. Revise the concentration of the compounds. Some of them are not in µg/L, as mentioned in lines 459-460, 462-463.

Response: Thank you for pointing it out, the concentration of the compounds in Table 3 have been revised.

Conclusions:

Lines 509-510: the aromas should be associated with the origins of the brandies, as described in Discussion (lines 444-446)

Response: Thank you for the suggestion and the sentence has been revised to “Results of sensory data and instrumental data showed that brandy from Yantai had a relative higher intensity on spicy and dried fruit, while brandies from Cognac had a relative higher intensity on floral and fruity.” at Lines 512-514.

LInes 511-512: the same approach should be used here. Associate the set of aromatic compounds with the brandies from China and Cognac, as described in Discussion (lines 480-484)

Response: Thank you for the suggestion and the sentence has been revised to “and the differences in their aroma characteristics could be associated with an aromatic balance between concentrations of a set of compounds such as 5-methylfurfural, γ-nonalactone, ethyl 2-hydroxy-4-methylpentanoate, ethyl lactate, and γ-dodecalactone.” at Lines 514-517.

Lines 512-514: once again the same approach should be used here. Associate the compounds derived from oak to VSOP and XO cognacs, as described in Discussion (lines 494-496) and showed in Figure 2d.

Response: Thank you for the suggestion and the sentence has been revised to “Meanwhile, VSOP brandy could be distinguished from XO brandy on the basis of compounds derived mostly during the aging process, such as ethyl vanillate, ethyl ac-etate, 2-acetylfuran, vanillin, and acetic acid.” at Lines 517-519.

Reviewer 3 Report

Comments and Suggestions for Authors

Dear Authors

the paper is an interesting  chemosensory characterisation of Chinese brandies and on their differences from Cognac.

It needs some main modifications.

Line 99: Table 1grade maybe type or ageing time?

Line 199-200: in which way the authors verified the normal olfactory abilities of the subjects? Did they do some tests?

Revise chapter 2.7 and separate the results and the methods.

Line 212-215: Put the results and the Table 2 in the chapter of Results.

Line 217-225: this sentence should be improved and put in the  chapter 3.Results, not in 2. Materials and Methods. 

In chapter 2, the authors can indicate, maybe in the 2.9 Data analysis: The sensory panel results were subjected to ANOVA (p=95%) also to verify the ability of the subjects.

In chapter 3 you can report these results:

The results of the ANOVA (Supplementary Figure 1) show the panel ability to differentiate the aroma descriptor (discrimination) consistently (repeatability) and consensually (agreement). There is a significant sample effect on all descriptors, and it is possible to conclude it can be concluded that the subjects were able to differentiate and correctly identify different odor descriptors.

Lines 264, 267,310, 321, and in all the paper: check the use of the word "grade"

Line 504: In this study, the aroma profile and odor-active compounds of different brandies with different quality grade from...

Line 512: Meanwhile, VSOP brandy with VSOP grade could be distinguished from XO brandy with XO grade on the basis of...

Minor revisions:

Line 14: what do you mean by grade? Maybe better quality grade 

Line 36:  "the main grade" please use a better word, it is not clear what the authors want to say, maybe the main types

Line 60-63: this sentence is not clear, please improve the sentence. Results showed that compounds could be specific to the origin of brandies... do you mean: Results showed that some specific compound could be related to the origin of brandies...?

Separate Line 355 and line 356 

Lines 356-377, 394-432, 437-514: these paragraphs should be justified.

Line 446: Our results...it they showed

Line 449-451: the aroma profile of Chinese brandies changed from This is not claer, what do you mean with from? attributes spicy to dried fruit, and the aroma profile of Cognac brandies changed from This is not claer, what do you mean with from? attribute alcohol and/or mushroom to fruity, floral and caramel.

Comments on the Quality of English Language

 The paper should be revised for the English, in some parts it is not clear waht the authors want to say.

Here are some suggestions, but please revise all the text.

Line 54: have been better were

LIne 59: also have been investigated. better were also been investigated

Line 80: has been used better was applied

Line 82: has been used better was employed

Line 161: EtOH/H2O better ethanol/water 

Line 431: For the “caramel” attribute, it showed that no compounds had...

Line 437, 439: have been performed better was performed

Line 437, 439: have been performed better was performed

Line 442: were have been used ...

Line 442-446: In Our result (Figure 1), showed that in a preliminary session eight aroma attributes described have been agreed in a preliminary session to describe the brandies from China and Cognac, and it showed that brandies from China had a relative higher intensity on of spicy and dried fruit odors, while brandies from Cognac had a relative higher intensity on of floral and fruity odors.

Line 453: the most intensity compounds presented in.. it is not correct, maybe 

the most relevant compounds found in..

Line 474: have been conducted... better were conducted 

Line 476-479: For freshly-distilled  French brandies within a limited geographic area, it is also possible to be distinguish different samples from an aromatic balance between concentrations of a set of common molecules such as ...this part is not clear, waht do you mean?

Author Response

Reviewer #3

Dear Authors

the paper is an interesting  chemosensory characterisation of Chinese brandies and on their differences from Cognac.

It needs some main modifications.

Line 99: Table 1grade maybe type or ageing time?

Response: Thank you for asking. After a careful literature review, we believe that “grade” is the most appropriate term to define the classification between samples, so we prefer to keep that term.

Line 199-200: in which way the authors verified the normal olfactory abilities of the assessors? Did they do some tests?

Response: Yes, all of the subjects went through screening tests that evaluated their performance in discriminating between different odors qualities and different odor intensity levels. The selection procedure of assessors was consistent with the flow in the GCO experiments at Lines 148-150.

Revise chapter 2.7 and separate the results and the methods.

Response: Thank you for the suggestion, this part has been modified.

Line 212-215: Put the results and the Table 2 in the chapter of Results.

Response: Table 2 has been moved to the chapter of Results.

Line 217-225: this sentence should be improved and put in the  chapter 3.Results, not in 2. Materials and Methods. 

Response: Thank you for the suggestion. This part has been modified and moved to chapter 3 at Lines 293-296.

In chapter 2, the authors can indicate, maybe in the 2.9 Data analysis: The sensory panel results were subjected to ANOVA (p=95%) also to verify the ability of the assessors.

Response: Thank you for the suggestion, the sentence has been revised to “The sensory panel results were subjected to analysis of variance (ANOVA, p = 95%) to verify the ability of the assessors with R software (version 4.0.1) by using the panelperf function from the SensoMineR package [23]” at Lines 242-244.

In chapter 3 you can report these results:

The results of the ANOVA (Supplementary Figure 1) show the panel ability to differentiate the aroma descriptor (discrimination) consistently (repeatability) and consensually (agreement)There is a significant sample effect on all descriptors, and it is possible to conclude it can be concluded that the assessors were able to differentiate and correctly identify different odor descriptors.

Response: Thank you for the comment, this sentence has been added at Lines 274-279.

Lines 264, 267,310, 321, and in all the paper: check the use of the word "grade"

Response: Thank you for the comment. After a careful literature review, we believe that “grade” was the most appropriate term to define the classification between the samples, so we prefer to keep that term.

Line 504: In this study, the aroma profile and odor-active compounds of different brandies with different quality grade from...

Response: Thank you for the suggestion, the word has been removed from the sentence.

Line 512: Meanwhile, VSOP brandy with VSOP grade could be distinguished from XO brandy with XO grade on the basis of...

Response: Thank you for the suggestion, the sentence has been modified. At Lines 517-518

Minor revisions:

Line 14: what do you mean by grade? Maybe better quality grade 

Response: Thank you for asking, the grade of brandy generally relies on a system of well-known acronyms that refer to the age of the youngest component in the blend. The definition of the term was given at Lines 44-49.

Line 36:  "the main grade" please use a better word, it is not clear what the authors want to say, maybe the main types

Response: Thank you for the comment. The sentence has been revised as “Brandies are usually aged in oak barrels and labeled according to their age, and the main labeling grade in the brandy market both in Cognac and in Yantai includes VS (very special), VSOP (very special old pale) and XO (extra old).” At Lines 44-46

Line 60-63: this sentence is not clear, please improve the sentence. Results showed that compounds could be specific to the origin of brandies... do you mean: Results showed that some specific compound could be related to the origin of brandies...?

Response: Thank you for the comment, it is the meaning that we want to explain. The sentence has been revised at Lines 67-68.

Separate Line 355 and line 356 

Response: The Line 355 (Now Line 349) belonged to Table 3 and format of the part has been modified.

Lines 356-377, 394-432, 437-514: these paragraphs should be justified.

Response: Thank you for the comment. According to all the reviewers’ suggestion, some sentences in these paragraphs have been modified and We have tracked each change by using a red font in the manuscript to make them easily available.

Line 446: Our results...it they showed

Response: The sentence has been revised to “Our results are consistent with that conclusion, which show (Figure 1) that as the grade of brandy increased, the aroma profile of Chinese brandies converted from spicy to dried fruit.” At Lines 446-448.

Line 449-451: the aroma profile of Chinese brandies changed from This is not clear, what do you mean with from? attributes spicy to dried fruit, and the aroma profile of Cognac brandies changed from This is not clear, what do you mean with from? attribute alcohol and/or mushroom to fruity, floral and caramel.

Response: Thank you for the comment, the sentence has been revised to “the aroma profile of Cognac brandies converted from attribute alcohol and/or mush-room to fruity, floral and caramel as the grade of brandy increased.” at Lines 453-455.

Round 2

Reviewer 3 Report

Comments and Suggestions for Authors

The paper can be accepted now.